# Plants and Their Derivatives as Promising Therapeutics for Sustainable Control of Honeybee (*Apis mellifera*) Pathogens

**DOI:** 10.3390/pathogens12101260

**Published:** 2023-10-19

**Authors:** Roberto Bava, Fabio Castagna, Stefano Ruga, Saverio Nucera, Rosamaria Caminiti, Maria Serra, Rosa Maria Bulotta, Carmine Lupia, Mariangela Marrelli, Filomena Conforti, Giancarlo Statti, Britti Domenico, Ernesto Palma

**Affiliations:** 1Department of Health Sciences, University of Catanzaro Magna Græcia, 88100 Catanzaro, Italy; roberto.bava@unicz.it (R.B.); rugast1@gmail.com (S.R.); saverio.nucera@hotmail.it (S.N.); rosamariacaminiti4@gmail.com (R.C.); maria.serra@studenti.unicz.it (M.S.); rosamaria.bulotta@gmail.com (R.M.B.); britti@unicz.it (B.D.); palma@unicz.it (E.P.); 2Mediterranean Ethnobotanical Conservatory, Sersale (CZ), 88054 Catanzaro, Italy; studiolupiacarmine@libero.it; 3National Ethnobotanical Conservatory, Castelluccio Superiore, 85040 Potenza, Italy; 4Department of Pharmacy, Health and Nutritional Sciences, University of Calabria, Rende, 87036 Cosenza, Italy; mariangela.marrelli@unical.it (M.M.); filomena.conforti@unical.it (F.C.); g.statti@unical.it (G.S.); 5Department of Health Sciences, Institute of Research for Food Safety & Health (IRC-FISH), University of Catanzaro Magna Græcia, 88100 Catanzaro, Italy; 6Nutramed S.c.a.r.l., Complesso Ninì Barbieri, Roccelletta di Borgia, 88021 Catanzaro, Italy

**Keywords:** *Apis mellifera*, honeybee pathogens, essential oils, plant extract, antiviral activity, antibacterial activity, antifungal activity

## Abstract

The most important pollinator for agricultural crops is the Western honeybee (*Apis mellifera*). During the winter and summer seasons, diseases and stresses of various kinds endanger honeybee numbers and production, resulting in expenses for beekeepers and detrimental effects on agriculture and ecosystems. Researchers are continually in search of therapies for honeybees using the resources of microbiology, molecular biology, and chemistry to combat diseases and improve the overall health of these important pollinating insects. Among the most investigated and most promising solutions are medicinal plants and their derivatives. The health of animals and their ability to fight disease can be supported by natural products (NPs) derived from living organisms such as plants and microbes. NPs contain substances that can reduce the effects of diseases by promoting immunity or directly suppressing pathogens, and parasites. This literature review summarises the advances that the scientific community has achieved over the years regarding veterinary treatments in beekeeping through the use of NPs. Their impact on the prevention and control of honeybee diseases is investigated both in trials that have been conducted in the laboratory and field studies.

## 1. Introduction

Since the honeybee (*Apis mellifera*), the most significant pollinator under commercial management, provides more than USD 200 billion in annual services to agriculture, honeybee diseases are a significant source of economic loss [1,2,3]. Although honeybees have developed effective defence mechanisms to counter infection at both the individual and societal levels, a variety of diseases and parasites can still be encountered in their colonies [4,5,6]. Infectious agents are known to spread more rapidly in overcrowded conditions. A large population of individuals within each colony seems to make honeybees particularly susceptible to pests and diseases. Some of the most significant infections affecting honeybees are the fungi *Ascosphaera apis* and the bacteria *Paenibacillus larvae* and *Nosema* spp (Figure 1). Mites also pose threats to *A. mellifera* [7,8,9]. Research has shown the crucial role of *Varroa destructor* (*V. destructor*) in the spread of viruses, which are responsible for significant losses of honeybee colonies in Europe and the United States [10,11,12]. Due to the considerable danger of contaminating apiary products, the use of veterinary chemical medications in the treatment and prevention of honeybees diseases has significantly decreased recently [13]. In order to effectively avoid the spread of diseases in honeybee colonies, it thus appears justified to pay closer attention to the value of preventive therapies and the search for new sustainable and safer remedies. For a very long time, herbal remedies have been utilised successfully in the treatment of human diseases [14,15]. New applications for their usage in human and veterinary medicine are simultaneously being found [16,17,18,19,20]. The usage of the majority of natural compounds (pyrethrum, neem oil, rotenone, nicotine) as insecticides dates back many years [21,22]. They typically decay faster in sunshine, air, and moisture than synthetic products, which is a major benefit [23]. They also do not bioaccumulate in the environment. They are often more selective against non-target insects as compared to conventional insecticides [23]. NPs consist of several biologically active substances with pharmacological action while causing fewer side effects than synthetic products [24,25]. One of their observed effects is that they directly or indirectly stimulate processes that increase the resistance and immunity of mammal and invertebrate organisms [26,27]. Several plant species have been identified so far as having immunostimulating characteristics [28,29]. A variety of plants and their derivatives also boost the body’s capacity to fight infections because of their energizing, regenerating, and metabolism-enhancing qualities [30,31,32]. In addition, some plants exhibit direct effects on pathogens by reducing their multiplication and consequently decreasing the infectious load [19,33,34]. Research on the potential application of plants and their components in beekeeping has so far been mostly directed at finding ways to utilise NPs to combat certain diseases by making use of their antibacterial, antifungal, or antiviral capabilities. The efficacy of herbal-derived medicines in treating European foulbrood (EFB), American foulbrood (AFB), and chalkbrood disease has been evaluated in several experiments. For example, essential oils (EOs) have been shown to be particularly effective against *Paenbacillus larvae* [35,36]. Several EOs, such as those obtained from oregano, fennel, mint, thyme, salvia, and pine oils, have all been used to try to manage the *V. destructor* pest [37,38,39]. This literature review came about with the aim of summarising the progress achieved in combating the main hive pathogens by means of NPs. Particular emphasis was given to the activity of EOs and their general mechanisms of action, concerning which there are a large number of scientific articles.

## 2. Mechanisms of Action

### 2.1. Antibacterial Activity

Given the many chemical constituents found in EOs, it is rare to be able to pinpoint a single mode of action against bacterial cells but rather a variety of targets against which these chemicals operate are commonly seen. Their hydrophobicity, which enables them to break the bonds between the lipids of the membrane of bacterial cells and mitochondria, is a crucial property of EOs and their compounds [40,41]. This property alters the structures of cell membranes that become more permeable as a consequence. The activity results in ions and other cellular components being lost. The cell can withstand some of these losses but often results in cell death [42,43]. According to Lambert et al., 2001, Gram-positive bacteria are often more vulnerable to the effects of Eos than Gram-negative bacteria [44]. The variations in cellular wall construction account for this finding. Gram-positive bacteria have thicker (about 20 nm) cell walls but simpler structures. They do not include lipids and are covered in a thick coating of peptidoglycan that is interspersed with linear chains of teichoic acid. Gram-negative bacteria have thinner cell membranes but more complex structures. They are made up of numerous peptidoglycan layers that are coated in a layer of lipoproteins and lipopolysaccharides that also include amino acids but lack teichoic acid [45]. The outer envelope is what increases the Gram-negative bacteria’s resistance to the effects of numerous chemicals, including those of plant extracts [46]. The crystal violet test and the measurement of the released UV-absorbing material assays may both be used to identify changes in membrane permeability [47,48]. The crystal violet test is based on the observation that when the cell membrane is damaged, the chemical is readily able to penetrate it. The idea behind UV-absorbing material tests is that Eos may damage cell membranes, allowing the cell’s contents to seep out and be detected on the UV spectrum. According to Denyer and Hugo, 1991, phenolic compounds like thymol, carvacrol, and geraniol have the most potent antibacterial effects [49,50]. These compounds work by altering the cytoplasmic membrane, disrupting the proton-motor force and electron flux, blocking active transport, and coagulating cell contents. The precise mechanism of the action and antibacterial activity of Eos are influenced by the chemical composition of their constituent parts [51]. Knobloch et al., 1986, and Dorman and Deans, 2000, have both proven the significance of a hydroxyl (OH) group in phenolic compounds like carvacrol and thymol [41,51]. In fact, the degree of antibacterial activity of thymol and carvacrol, both of which have the hydroxyl group at distinct positions, has been revealed to be very comparable for bacteria like *Bacillus cereus*, *Staphylococcus aureus*, and *Pseudomonas aeruginosa* [44,52]. Carvacrol and thymol, however, have been shown to operate against Gram-positive and Gram-negative bacteria differently in other experiments [51]. The ineffectiveness of menthol, which lacks the phenolic ring, highlights the significance of its existence. Instead, the inclusion of the acetate group seems to improve the antibacterial activity; in fact, geranyl acetate appears to be more effective than geraniol against various species of Gram-positive and Gram-negative bacteria [51]. In addition, it has been possible to highlight that the biological activity of the nonphenolic components that make up the EOs may differ depending on the type of functional group that is present; for instance, the alkenyl group (C = C) is more efficient than the alkyl group (C-C), making limonene more active than p-cynene [51]. According to Knobloch et al., 1989, the constituents of EOs seem to have the ability to interact with the proteins that make up the cytoplasmic membrane [40]. It is common knowledge that lipid molecules surround membrane-bound enzymes like ATPase, and two potential mechanisms of action have been proposed: first, when a cyclic hydrocarbon acts on these molecules, lipophilic molecules may accumulate in the membrane’s lipid bilayer, changing the way proteins and lipids bind to one another; second, a direct change within the lipophilic part and the hydrophobic part of the protein may occur [50,53]. Therefore, the various EOs mechanisms of action are diverse and distinct from one another, resulting in a sum of every mechanism of action of their individual component. Because of this high level of variability, it is believed that microorganisms cannot develop resistance to these substances [6,42]. However, variations in composition among different Eos, even from the same kind of plant, may pose a problem as their antibacterial actions could vary. For the evaluation of the activity, it is necessary to remark that there are currently no accepted standards for classifying the antibacterial activity of NPs. According to Holetz et al., 2002, the antibacterial activity of vegetal extracts was excellent if the minimum inhibitory concentration (MIC) was less than 100 µg mL^−1^; moderate if it was between 100 and 500 µg mL^−1^; weak if it was between 500 and 1000 µg mL^−1^; and inactive if it was above 1000 µg mL^−1^ [54].

### 2.2. Antifungal Activity

EOs have the potential to disrupt, modify, or impede the production of cell walls in fungus. Anise oil’s main ingredient, trans-anethole, showed antifungal action against the filamentous fungus *Mucor mucedo* IFO 7684, along with morphological alterations to the hyphae, such as enlarged tips. In permeabilised hyphae, anethole dose-dependently reduced chitin synthase (CHS) activity [55]. *Aspergillus niger* cannot develop when exposed to the EO from *Citrus sinensis* epicarp, which has a high percentage of limonene (84.2%). It also causes permanent negative morphological changes, including the depletion of cytoplasm in the hyphae of fungal organisms and the budding of hyphal tips [56]. The main chemical in the EO of black cumin seed, thymoquinone, caused significant fungal cell wall and cell membrane damage, according to Iscan et al., 2016 [57]. A large proportion of phenolic monoterpenes, including thymol and carvacrol, was present in certain EOs, while others contained mostly non-phenolic terpenes. In EOs with potent efficacy against mould, the phenylpropanoid eugenol was often found. The bicyclic monoterpenes camphor and pinene, which are non-phenolic, have shown notable antifungal effects in certain instances. High thymol, carvacrol, cymene, linalool, or α-pinene concentration has been seen in EOs with potent action against yeasts. Surprisingly, a number of EOs with inhibitory activities against yeasts were discovered in the Lamiaceae family of genera, including *Thymus* spp., *Origanum* spp., Salvia Rosmarinus, *Ocimum sanctum*, and *Zataria multiflora* [58,59]. *Citrus sinensis* EO was found to be abundant in limonene (84.2%). It displays a strong antifungal effect against *Aspergillus niger*, causing the rupture of the mycelial walls [56].

Furthermore, it has been shown that EOs may stop spores from growing. Sant’Anna et al., 2009, found that *Achillea millefolium* EO, containing 42.2% chamazulene, has genotoxic effects on fungal cells and inhibited spore growth [60].

By preventing the activity of mitochondrial dehydrogenases involved in ATP generation, such as lactate dehydrogenase, malate dehydrogenase, and succinate dehydrogenase, some EOs may impact the efficiency of the mitochondria. According to Chen et al.’s research, 2013, the EO of *Anethum graveolens* may serve as an antifungal agent in addition to disrupting the citric acid cycle and preventing the generation of ATP in *C. albicans* mitochondria [61].

Another mechanism by which EOs act is by interfering with the membranous pump flows of fungi. The enormous transmembrane electrochemical proton gradient required for food intake is supported by the fungal plasma membrane H^+^-ATPase, which is crucial to the fungal cell’s function. In addition to controlling internal pH and fungus cell development, the H^+^-ATPase also affects dimorphism, nutrient absorption, and medium acidification, which all contribute to fungus pathogenicity [62,63]. Cell death and intracellular acidification result from H^+^-ATPase inhibition. This is the target of eugenol and thymol, which are fungicidal agents even against azole-resistant fungi [64]. The main chemical components of thyme EO, thymol and carvacrol, combined with the azole antimycotic fluconazole showed an antifungal effect that was synergistic. This was accomplished by preventing the overexpression of the efflux-pump genes CDR1 and MDR1 in *Candida albicans* [62].

### 2.3. Antiviral Activity

Viral attachment, penetration, and entry into the host cell result in the replication of genetic material and the production of additional virions. Viral infectivity is inhibited by adverse action on associated targets engaged in the infection phase. The greatest options for clinically effective antiviral drugs are those that target certain viral biosynthetic stages. These antiviral formulations should stop the propagation of viruses, work at low concentrations, have no effect on the functioning of the host cell, and eventually heal infected cells. The main mechanism of antiviral effects of EOs is the capsid disintegration, which prevents the virus from infecting host cells [65]. Being lipophilic in nature, EOs can penetrate viral membranes, easily leading to membrane disintegration. EOs also inhibit the haemagglutinin of certain viruses; this membrane protein allows the virus to enter the host cell [66]. Many EOs and their components can inhibit the late stages of the viral life cycle by targeting the redox signalling pathway [67]. EOs of *Thymus vulgaris*, *Cymbopogon citratus* and Salvia Rosmarinus have been found to destabilise the Tat/TAR-RNA complex of HIV-1 virus, an essential complex for HIV-1 replication [68]. EOs have been tested against a large number of pathogenic viruses, such as influenza and other respiratory viral diseases [66,69]. According to studies, the EOs of cinnamon, bergamot, lemongrass, thyme, and lavender have strong antiviral activity against influenza A virus [70]. Eos of *Citrus reshni* leaves has been shown to have beneficial effects against the H5N1 virus [71]. In Vero cells, it has been shown that the EO of the *Lippia* species at a concentration of 11.1 µg/mL causes the 100% inhibition of the yellow fever virus [72]. Through in vitro and in vivo tests, it has been shown that oleoresins and EOs may have antiviral effects against the infectious bronchitis virus caused by the avian coronavirus [73]. This evidence supports the possible use of EOs as antiviral agents in beekeeping.

### 2.4. Acaricidal Activity

Some fixed (solvent) extracts from plants have been investigated as potential varroacidal sources (as mixtures or isolated compounds) [74,75,76,77]. However, EOs are among the natural substances that have been most thoroughly investigated for this use. As a statement of reality, searching for EOs with anti-varroa activity represents a logical strategy as EOs show several properties against arthropods, such as repellence [78], dissuasion [79], and toxic effects when fumigated, externally applied, or eaten [80,81]. According to Isman, 2000, toxicity may result in cuticle disruption, moulting, and respiration suppression, as well as a decrease in growth and fecundity rates [22]. These diverse forms of pharmacological action may be caused, at the molecular level, by the inhibition of acetylcholinesterase. However, this mechanism was demonstrated only in vitro for a small subset of monoterpenes [82]. Investigations into the mechanisms of action of monoterpenoids suggest that a mode of action for EOs is the inhibition of acetylcholinesterase enzyme activity. *Thymus praecox* subsp. *caucasicus*, *Cyclotrichium niveum*, *Santolina chamaecyparissus*, *Ormenis multicaulis*, *Echinacea purpurea*, *Salvia chionantha*, *Anethum graveolens*, and *Salvia lavendulaefolia* have all been shown to inhibit AChE [83,84,85,86,87]. Multiple aspects of this inhibitory activity were assessed. Among 73 substances, 48 that showed anti-AChE activity were assessed. When 28 compounds were tested on insect AChE, 23 of them inhibited the enzyme. 1,8-cineole, cis-ocimene, niloticin, limonene, menthol, α-pinene, β-phellandrene, and carvacrol were the most efficient [86,87,88,89,90]. The bulk of the EO components exhibited anti-AChE activity at mM doses. Only one investigation demonstrated that the carvacrol component of EOs may inhibit AChE in a µM concentration [91].

The majority of studies have linked the biological activity of EOs to the prevention of octopamine (OA)-binding to their receptors [92,93,94], as there are parallels between OA activity and exposure to monoterpenoids as well as evidence that OA antagonists block the effects of monoterpenes on insect neuronal responses [92,95]. The final effect is a decrease in the concentration of cAMP. This mechanism has the benefit that OA is not employed by vertebrates as often as it is by arthropods [96], allowing selection between these species to be at least theoretically possible when searching for biopesticides to be used against arthropods. Numerous EO components have been shown to have pharmacological effects and affect insects’ octopaminergic systems [97]. For example, geraniol and citral greatly reduced cAMP levels. The same EOs decreased [3H] OA’s affinity for receptors. Another mechanism by which EOs function is via GABA-gated chloride channels [98]. Since insect GABArs differ from human GABArs in both structure and pharmacology; these are very intriguing targets for the creation of novel pesticides. Thymol, menthol, and other substances increase the Cl- current brought on by the GABA neurotransmitter. The GABAArs Cl- current is unaffected by several EO elements, including camphor, carvone, menthone, linalool, and α-terpineol. The chemical makeup of the EO components affects how they interact with GABA receptors. The ability of various EO stereoisomers to modulate GABA receptors varies; (+)-menthol and (+)-borneol are more potent than (−)-menthol and (−)-borneol. Likewise, the functional group is also essential. The GABAArs are more modulated by alcohols like thymol, menthol, and borneol than by ketones like linalool and α-terpineol.

## 3. Control of Honeybee Pathogens and Parasites via NPs

The hive superorganism has to deal with numerous pests and diseases. Some of them are minor and the diseases they produce can be managed even after they have spread. They will therefore not threaten the survival of the colony. Others can lead to colony collapse if they go unnoticed. In the following subsections, we will take stock of the achievements of the scientific community in controlling important bacterial, fungal, viral, and parasitic bee diseases using NPs.

### 3.1. Paenibacillus larvae and Melissococcus plutonius

AFB is the most serious and pervasive infectious disease of honeybees and affects the brood. It is present in practically every nation on every continent, with the exception of a few tropical regions in Africa and Asia [99,100]. For this reason, it causes huge financial losses in beekeeping worldwide [101]. A Gram-positive, facultative anaerobic, sporigenic bacteria known as *Paenibacillus larvae*—previously known as *Bacillus larvae*—is the culprit [99]. The bacteria originate from the spores that, in a favourable environment and under ideal circumstances, may germinate and multiply in approximately 30 min [102]. The spores are the bacterium’s resistant form because they are covered in a very durable membrane that shields them from environmental stresses. The digestive system of immature honeybee larvae provides the semi-aerobic environment that spores need to germinate. Therefore, after being consumed with food, the spores enter the lumen of the midgut where they germinate. Here, the vegetative bacterial forms begin to grow. Massive bacterial growth occurs at the expense of the food consumed by the larva. It is only later that the bacteria breach the peritrophic membrane, assault the intestinal epithelium, move into the haemocele, and grow in the haemolymph of the larva, resulting in septicaemia that leads to larval death [103,104].

AFB has no effective treatments at this time. This disease has been treated with antibiotics with predominantly bacteriostatic activity, such as oxytetracycline hydrochloride [105,106]. If used inaccurately, they mask the symptoms of the disease but not its spread. Furthermore, drug resistance phenomena could occur. For these reasons and the possibility of contaminating hive products, several countries prohibit the use of antibiotics in the treatment and prevention of AFB [13]. It is also important to remember that unlike the vegetative form, the spore-forming form cannot be affected by antibiotics or chemotherapeutics. The best course of action in situations of overt sickness is still to destroy infected colonies and combs by burning them. AFB prevention and control techniques often involve monitoring for early diagnosis, isolating apiaries with instances of AFB, expanding healthy colonies using hygienic queens, and practicing the shaking technique when symptoms are already present. As far as European foulbrood (EFB) disease is concerned, it should be said that it is caused by the bacterium *Melissococcus plutonius* (*M. plutonius*), which is often associated with other bacterial agents, including *Streptococcus faecalis*, *Achromobacter eurydice*, *Paenibacillus alvei*, and *Bacillus laterosporus* [107,108]. Depending on the type of bacteria associated with *M. plutonius*, European foulbrood may manifest itself in the affected family with a different symptomatology (e.g., the presence/absence of particular odours). *M. plutonius* is a germ that, although asporigenic, is fairly resistant to environmental adversity: it can resist desiccation for up to a year; it can remain viable in pollen for several months. Similarly to AFB, within the hive, the disease is spread orally by nurse honeybees, which unwittingly smear themselves with germs in an attempt to clean the brood cells of dead EFB larvae and then, when they feed the brood, they infect them [108,109]. From hive to hive or from apiary to apiary, the disease can be spread either through honeybee action (especially when looting takes place) or by mistakes made by the beekeeper (using infected honey to feed healthy families, moving sick families during nomadism, trading in infected beekeeping material, using contaminated equipment, moving combs from one hive to another, etc.) [108,110,111]. Although the disease can occur at any time of year, it is more frequent in spring/summer, when the brood is at its peak. This observation led to the assumption that the disease could be favored by an imbalance between the number of larvae and the number of nurse honeybees. Furthermore, EFB appears to be more frequent in cold and rainy springs, when there may be food shortages, particularly of protein, as the brood lacks pollen [112]. After infection, the larvae reach death within the first four days of life, (regardless of whether the larvae are worker, drone, or queen larvae). The death of the larvae therefore takes place in an open cell, and this is one of the characteristics that allows EFB to be differentiated from AFB [110,113]. Prophylaxis involves renewing combs every 2–3 years, eliminating queens predisposed to the disease, carrying out artificial swarming correctly, and trying to avoid numerical imbalances between adult honeybees and brood. As with AFB, in conjunction with other good beekeeping practices, the shook swarm procedure may prove useful in suppressing infection [114,115]. In this particular context, the creation of innovative and efficient techniques for the management and prevention of AFB and EFB disease is critical. These techniques may take into account proof of bacterial resistance phenomena, adhere to tight EU requirements, and reflect current green consumption patterns [116,117]. Alternative methods of preventing and treating AFB and EFB are now being researched, with a focus on EOs, propolis and probiotics [118,119]. The capacity of certain EOs to stop *P. larvae* from growing was examined in a number of experiments that are presented below. In the studies of Alippi et al., 1996 and 2001, the antimicrobial effects of EOs on *Paenibacillus larvae* were investigated using summer savory (*Satureja hortensis*), lavender (*Lavandula hybrida*), eucalyptus (*Eucalyptus globulus*), lemongrass (*Cymbopogon citratus*), peppermint (*Mentha piperita*), oregano (*Origanum vulgare*), rosemary (Salvia Rosmarinus), and thyme (*Thymus vulgaris*). The strongest action against *P. larvae* was found in EO of lemongrass, and thyme [35,120]. Similar positive conclusions were reached by Fuselli et al., 2005 and 2006, verifying that Andean thyme (*Acantholippia seriphioides*) had a marked antibacterial effect [36,121]. This EO demonstrated the highest inhibitory activity against the agent responsible for AFB. Kloucek et al., 2008, showed that, under the tested concentrations of 1–256 nL/mL, the EO of *Armoracia rusticana* (MIC 16 nl/mL) had the most pronounced impact on *Paenibacillus larvae*, followed by oils of *Thymus vulgaris* (MIC 64 nl/mL), *Mentha spicata* var. *crispa* (MIC 64 nl/mL), and *Satureja hortensis* (MIC 128 nl/mL), which, however, showed moderate efficacy [122]. In 2009, Gende et al. demonstrated that the EO of cinnamon (*Cinnamomum zeylanicum*) possesses AFB disease control activity similar to that of oxytetracycline [123]. Both a lab test and a field experiment were conducted to determine whether the EO of cinnamon had anti-*Paenibacillus larvae* activity. The tube dilution technique was used to estimate the MICs (minimum inhibitory concentrations) against *P. larvae*. The apiary study was conducted employing three groups of five hives each. Three doses of oxytracycline-HCl (0.4 g each) were given to the first group, two doses of cinnamon oil (1000 g/mL each) were given to the second group, and the third group was kept untreated as a control. Treatment effectiveness was deducted, observing a 360 cm^2^ (18 × 20 cm) brood surface and the number of infected brood cells on both sides of a central comb. The *C. zeylanicum* MICs for the antibiotic and EO were 50 µg/mL and 3.125 µg/mL, respectively, for the Mar del Plata AFB strain. Control beehives showed more infected cells than treated beehives. The antibiotics and the cinnamon EO, which were both tested, were thus successful in controlling AFB [123]. In 2012, Cecotti et al. demonstrated cinnamon-like efficacy for the volatile fraction of *Polygonum bistorta*. They also added an additional important piece of information, namely that plant activity varied with phenological stage [124]. *Melissococcus plutonius* and *Paenibacillus larvae* were used as test subjects for the EO. The inhibition area in the tests against *Paenibacillus larvae* ranged from 7.5 to 8.5 mm for the vegetative phase, from 7.5 to 13.0 mm for the blooming phase, and from 24.0 to 24.5 mm for the fruiting phase, demonstrating a noticeably rising level of activity over time. In the test against the three strains of *Melissococcus plutonius*, absolute inhibition levels were less prominent, but the results revealed a similar upward trend. Data from the bark and leaves of *C. zeylanicumin*, one of the botanical species that is most effective against the AFB, were compared with those from the isolates in order to provide a better benchmark for the amount of bioactivity of the isolates. *P. bistorta* oils demonstrated antibacterial activity against all bacterial strains with areas of inhibition that varied from 2.5 mm toward *M. plutonius* CRA-API08/1 to 24.5 mm toward *P. larvae* CRA-API10/8, showing inhibition values similar to those of *C. zeylanicum* in the flowering phases. It is important to note that *P. bistorta* in the fruiting phase showed inhibition values similar to those displayed by oxytetracyclin against *P. larvae* (36.0–39.0 mm inhibition area). While EOs from rosemary, and fennel exhibited minimal or poor antibacterial action, Kuzysinova et al., 2014, identified the maximum suppression of *P. larvae* using EOs of oregano, thyme, and clove [125]. In a more recent study, Pellegrini et al., 2017, examined the EOs of *Aloysia polystachya*, *Acantholippia seriphioides*, *Schinus molle*, *Solidago chilensis*, *Lippia turbinata*, *Minthostachys mollis*, *Buddleja globosa*, and *Baccharis latifolia* for their antibacterial activity [126]. At 260 nm, EO-induced releases of UV-absorbing *P. larvae* cell material were detected. With the exception of *B. latifolia* EO, all EOs exhibited antibacterial action against *P. larvae* and damaged the cell wall and cytoplasmic membrane of *P. larvae*, causing the leaking of cytoplasmic components [126]. Using the agar diffusion method, the antibacterial effects of several extracts from *Flourensia riparia*, *Flourensia fiebrigii*, and *Flourensia tortuosa* against *P. larvae* were also studied. Using this method, the extracts of chloroform, ethyl ether, and hexane at varying concentrations (100 to 50,000 µg mL^−1^) exhibited an inhibitory effect [127]. One hundred micrograms of dry extracts isolated from the inflorescences of different *Hypericum* species showed similar antibacterial action against *P. larvae*. The diameters of the inhibitory halos were similar when applying as little as 25 µg of these extracts when compared to 1000 µg of lactic acid (positive control) [128]. The antibacterial effects of *Vitex trifolia* (Barbaka) and *Azadirachta indica* (Neem) crude aqueous extracts (20, 40, and 60 mg mL^−1^ disk^−1^) against *P. larvae* were dose-dependent, with the larger inhibition zones being associated with the Barbaka plant [129]. According to Gonzaléz and Marioli, 2010, water extracts of *Achyrocline satureioides*, *Chenopodium ambrosioide*, *Eucalyptus cinerea*, *Gnaphalium gaudichaudianum*, *Lippia turbinata*, *Marrubium vulgare*, *Minthostachys verticillata*, *Origanum vulgare*, *Thymus vulgaris*, and *Tagetes minuta* prevented the development of *P. larvae* [130]. The several vegetal species under study varied in their biological activity against the development of *P. larvae*. *E. cinerea*, and *M. verticillata* were those with the greatest activity, having completely prevented the development of any strain of *P. larvae*. However, the growth of the *P. larvae* strains was also likewise inhibited by the *T. minuta* extracts with significant effectiveness (91%). Another plant whose extracts displayed biological action as *P. larvae* growth inhibitors was *A. satureioides* (83% effectiveness). The plant species whose extracts were least effective in inhibiting *P. larvae* growth were *O. vulgare* and *T. vulgaris*. While *T. vulgaris* displayed 50% effectiveness, *O. vulgare* only displayed 42% [130]. By using the broth microdilution and agar dilution procedures, the hexane extract of *A. satureioides*’ MIC values were found. These values ranged from 16 to 125 µg mL^−1^ [131]. In 2014, other extracts of the same plant were analysed using the broth microdilution technique. The hexane, benzene, ethyl ether, and ethyl acetate extracts of *A. satureioides* produced MIC values of 0.060, 0.131, 0.773, and 6545 µg mL^−1^, respectively [132]. According to Gende et al., 2008, the broth macrodilution technique yielded a MIC of 5000 µg mL^−1^ for the ethanolic extract of *Melia azedarach* [133]. The ethanol–water extracts of *Calendula officinalis*, *Nasturtium officinale*, and *Laurus nobilis*, as well as the crude extracts of *Scutia buxifolia* and its extracts extracted with dichloromethane, ethyl acetate, and n-butanol fractions were also effective [117,134,135]. A recent study, conducted by Kim et al., 2018, evaluated the antibacterial activity of several molecules isolated from plant seeds and fruits (macelignan, tracheloside, fangchinoline, corosolic acid, loganic acid, emodin-8-O-β-D-glucopyranoside, and dehydrocostus lactone) against the causal agents of European and American foulbrood, *P. larvae*, and *Mellisococcus plutonius* [136]. Using the broth microdilution assay, the authors discovered that macelignan possessed the highest degree of anti-*P. larvae* activity (MIC of 1.56 mg/L after a 24 h incubation and MIC of 3.125 mg/L after a 48 h incubation), followed by corosolic acid, which had similar anti-*P. larvae* activity (MIC of 3.125 mg/L after a 24 h and 48 h incubation). The anti-*M. plutonius* activity of macelignan and corosolic acid was similarly quite strong (MIC of 3.125 mg/L at 24 and 48 h of incubation). Loganic acid, tracheloside, fangchinoline, dehydrocostus lactone, emodin-8-O-D-glucopyranoside, kanamycin, and Congo Red demonstrated little or no antibacterial activity, whereas miconazole (1.56~3.125 mg/L) and tetracycline (0.39~1.56 mg/L) proved high antibacterial action against the two strains of bacteria [136]. Hassona confirmed in 2017 that some botanical species have entirely distinct efficacies for AFB and EFB. In both the laboratory and the field, the efficacy of *Cinnamomum zeylanicum*, *Syzygium aromaticum*, and *Thymus vulgaris* against AFB and EFB bacteria was assessed. Thymol had the greatest impact on AFB in the lab, with a total average of 3.37 ± 1.03 cm of growth suppression measured as a circle around the spreading disc for each concentration. Thymol, on the other hand, had the least impact on EFB, with an overall mean of 0.33 ± 0.15 cm. With a total mean of 4.50 ± 2.00 cm and 0.73 ± 0.20 cm on EFB, cloves had the greatest impact. Thymol produced the best brood growth in the apiary, with a 25.8% increase in capped brood over the entire period of treatment. With a total mean of 4.50 ± 2.00 cm and 0.73 ± 0.20 cm on EFB, cloves had the greatest impact. In the apiary, thymol produced the greatest levels of brood growth, with a 25.8% increase in capped brood compared to EFB throughout the course of the treatment period. Cloves, on the other hand, varied the most across each control period, with a 25.3% rise in capped brood across all treatment times [137].

However, the possible adverse effects of EOs on honeybees should not be overlooked when examining the efficacy of these compounds against infections. Kevan et al., 1999, in addition to determining the effectiveness of many EOs in slowing down the development of *P. larvae*, also provided the LD50 values of the relevant extracts for honeybees [138]. LD50 values for thymol reached 100 ppm, for cinnamon oil 50 ppm, and for clove oil 200 ppm, while peppermint oil was found to be absolutely non-toxic. The LD50 values of thyme, lemongrass, oregano, and basil extracts for adult honeybees were established by Albo et al., 2003 [139]. For the pure essences mentioned above, the LD50 could not be determined, as it yielded a negative curve for the mortality values. The absent or low toxicity to honeybees found in these trials support the use of these EOs/extracts in honeybees.

### 3.2. Nosema spp.

Adult *A. mellifera* are susceptible to the infectious disease nosemiasis, which is brought on by the unicellular fungus of the class microsporidia. These are eukaryotic unicellular organisms that reproduce at the cost of the adult honeybees’ gut epithelial cells. Depending on the *Nosema* species involved, the breed of bee, and the environmental circumstances in which the bees are situated, the disease presents clinically in various ways. There are two main species of *Nosema* that affect honeybees: *N. apis* and *N. ceranae*. The first causes a typical digestive syndrome whose main symptom is diarrhoea. The second has no particular signs, but manifests as a depopulation of the hive [140]. Spores, which are *Nosema* in its latent condition, are the disease’s main means of transmission. They are surrounded by a thick envelope produced by the cell contained in it; the spore also contains a polar filament, which departs from a sac, known as “polar tube”, that surrounds the peripheral part of the cell [141]. When spores consumed with food make it to an adult bee’s gut, they germinate and produce an amoeboid form with cytoplasm devoid of mitochondria. The filament guarantees adhesion to host cells and acts as an exit duct. The amoeboid form enters the intestinal cells where it starts to grow, develop, and multiply by feeding on the cytoplasm of the cell. In this location, they grow and multiply, producing a large number of cells, each of which will give rise to a spore until the invading cell explodes. The mature spores are then released into the intestine. The spores, which represent the spreading forms of the disease, are poured into the lumen of the intestine and there expelled with the faeces when the intestinal epithelium is regenerated. Upon reaching the outdoor environment, the spores can be ingested by other honeybees, and the biological cycle closes (orofaecal route) [140].

Until a few years ago, Flumidil B was used to control this microsporidian parasite. This antibiotic has been banned in all European Union member states since 2012 and in Italy since 2002. Numerous studies have been carried out on the effects of adding natural derivatives to the diet of honeybees and the prevalence of *Nosema* spp. cases. In a 2008 study, Maistrello et al. assessed the efficacy of lysozyme, resveratrol, thymol, and vetiver EOs in the treatment of honeybee nosema infection [142]. The experimentally infected groups received candies made with thymol (0.12 mg/g), lysozyme (2.5 mg/g), vetiver (1.2 mg/g), and resveratrol (0.01 mg/g). Spore loads were observed for 25 days. The findings demonstrated that vetiver oil, thymol, resveratrol, and lysozyme had neither harmful nor anti-feedant effects on adult honeybees, since candy consumption was comparable throughout the experimental groups. The study revealed that honeybees given thymol and resveratrol candies had much-reduced infection rates. Furthermore, the resveratrol group survived noticeably longer than honeybees that were fed candy without resveratrol [142]. Thymol had similar impacts on the health of *Nosema*-infected honeybees, according to Glavinic et al., 2022, who discovered elevated levels of immune-related genes and oxidative stress parameter values as well as a reduction in *Nosema* spore burden. Thymol may create issues in honeybees not afflicted by *Nosema* (affecting bee survival, lowering oxidative capacity, and downregulating several immune-related gene expressions), according to the same investigators, who also recommended careful, non-preventive usage of the substance [143]. The EO of *Cryptocarya alba* was also tested. According to a study by Bravo et al., 2017, *C. alba* includes 39 different chemicals and, among them, the three main ingredients α-terpineol, eucalyptol, and the monoterpene α-phellandrene. In the study, infected honeybees were divided into three groups: those who received EO *C. alba* at various dosages (1, 2, 3, and 4 µg/bee); those who received fumagillin syrup (240 µg/bee) as a positive control; and those who were infected and were maintained without any treatment. The findings revealed that the most efficient dosage of 4 µg EO/bee had a spore inhibition rate of 80%, which was comparable to fumagillin. Furthermore, the EO was not harmful to *A. mellifera* [144]. EOs from a several number of plants, including peppermint (*Mentha piperita*), lemon balm (*Melissa officinalis*), summer savory (*Satureja hortensis*), and coriander (*Coriander sativum*), have shown anti-nosemosis efficacy and extended the lifespan of diseased honeybees [145]. The product “Supresor 1” (a combination of the abovementioned medicinal herbs extracted using ethyl alcohol) was given to six groups of five honeybees each (experimental modules) at various concentrations of 1 mL, 2 mL, 5 mL, 10 mL, and 50 mL per litre of syrup, along with a positive control (infected non-treated) and two negative control groups (uninfected, treated). Even at a high dosage of 10 mL (2000 mg etheric oil) per litre of syrup, the results showed no toxicity towards honeybees; the ideal dose found was 5 mL per litre of sugar syrup [145]. Plant extracts have also been used to control nosemiasis. As shown by a 77% decrease in spores at a concentration of 100 µg/mL, the ethanolic extracts of *Artemisia dubia* and *Aster scaber* exhibit anti-nosemosis properties [146]. Surprisingly, the spore levels were reduced by 76% using aqueous extracts of 1 µg/mL. Aqueous extracts were more active than butanol and ethyl acetate extracts [147]. Six adaptogenic plant extracts, including those from *Eleutherococcus senticosus*, *Garcinia cambogia*, *Camellia sinensis*, and *Schisandra chinensis*, were examined. The highest anti-nosema action was seen in an extract of the *E. senticosus* root [148]. A second investigation demonstrated the anti-nosemosis effectiveness of aqueous nest carton extracts made from a jet-black ant nest (*Lasius fuliginosus*) with no harm to healthy bees. The amount of spores was also reduced by 97.97% by the birch carton extract [149]. The aqueous extract of the *Agaricus blazei* mushroom proved successful in lowering the amount of *N. ceranae* spores without creating any negative side effects when used either at the time of infection or as a prophylactic strategy. Most immune-related genes, including abaecin, hymenoptaecin, defensin, and vitellogenin, were expressed more often in the presence or absence of *Nosema* infection thanks to *A. blazei*. The extract was well tolerated and acceptable, as seen by the fact that daily dietary intake was similar among the groups [150,151]. In another experiment, in order to feed the infected honeybees, defatted seed meals (DSMs) from *Brassica nigra* and *Eruca sativa* were administered for 8 days together with a known number of different glucosinolates. The amounts, which were 2% and 4%, resulted in the suppression of *N. ceranae* as well as possible nutritional advantages as shown in the length of the honeybees’ lives [152]. *Aristotelia chilensis*, *Ugni molinae*, and *Gevuina avellana* leaves, as well as propolis, were tested for their effects on *N. ceranae* infection. Propolis and *U. molinae* extracts, which were successful in treating infected groups, were determined to be adequate [153]. Another investigation on the impact of ethanolic propolis extract on the survival and spore load of worker honeybees infected with *N. ceranae* revealed that propolis significantly decreased the *Nosema* spore load in comparison to the control [154]. Furthermore, honeybees with *N. ceranae* infections lived longer when given propolis extracts and ethanol (a solvent control), but only propolis extract effectively decreased spore burden [155]. The mortality, infectivity, and *N. ceranae* infection rates were all considerably lower in honeybees that were administered propolis ethanol extract either before or after infection than in the positive control [156]. At the conclusion of this section, it should be noted that there are already licensed and commercially available plant essence products for the control of nosemiasis. Their efficacy has been proven in experimental studies. For example, a statistically significant decrease in the amount of *N. ceranae* spores was seen when the herbal supplements Nozemat Herb^®^ and Nozemat Herb Plus^®^ were used [59]. Additionally, employed as anti-*N. ceranae* therapy were the dietary supplements ApiHerb^®^ and Api-Bioxal^®^. Both treatments reduced the frequency of infections and the quantity of *N. ceranae*, but ApiHerb^®^ had a greater effect [157]. Therefore, with the provision of commercially available supplemental feeds (Apiherb^®^, Vita Feed Gold^®^), honeybee colonies may be energised and strengthened.

### 3.3. Ascosphaera apis

The fungus ascomycete *Ascosphaera apis* (*A. apis*) is the cause of chalkbrood disease or “ascospherosis”. Larvae of all castes can be affected, but there is a higher initial prevalence in drones. This usual spring disease weakens the colony by lowering its production and exposing it to additional diseases, such EFB. It seldom results in the mortality of the infected colony. Due to the lack of bacterial competition from various species of Bacilli and Penicilli that are often present in healthy hives, the chalkbrood disease may be observed more frequently in antibiotic-treated colonies [158]. Another cause underlying the occurrence of the disease can be found in incorrect thermoregulation. Due to the decreased heating capacity of the brood region, weak nuclei and colonies are the populations that are most in danger. For the same reason, drone larvae often found on the periphery of the brood chamber are the most affected. If the hive is densely occupied, honeybees will respond to the fungus by increasing the temperature within the nest (social fever) [159]. Regarding pathogenesis, larvae become infected by consuming *A. apis* spores dispersed in the environment along with food during their first few days of life (usually between the third and fourth day). These sprout in the intestinal lumen, most likely stimulated by the elevated CO_2_ content. The mycelium then starts to develop and breaks through the intestine’s walls, spreading throughout the rest of the body, often from the back to the front. In most cases, larvae die within two days after wax layer apposition or soon after. The larva then experiences dehydration, which causes calcification/mummification. The consistency changes from soft to rubbery to hard, as if calcified, over time, giving rise to the disease’s name. A vast array of products has been investigated for the control of the chalkbrood disease and, among them, many EOs have been tested. Gochnauer and Margetts’ 1979 study, one of the first, demonstrated via in vitro testing how citral and geraniol suppressed the development of the fungus *A. apis* by blocking its vegetative growth [160]. The antifungal properties of natural substances were still being studied by Calderone et al. in 1994. In particular, the authors evaluated eight plant extracts: bay oil (*Pimenta racemosa*), camphor, clove oil (*Syzygium aromaticum*), cinnamon oil (*Cinnamomum* sp.), citronellal (3,7-dimethyl-6-octenal), Spanish origanum oil (*Thymus capitatus*), α-terpinene (l-isopropyl-4-methyl-1,3-cyclohexadiene), and thymol (5-methyl-2-[l-methylethyl]phenol). At 100 ppm for 168 h, cinnamon oil completely prevented fungal proliferation. For 72 h at 100 ppm and for 168 h at 1000 ppm, origanum oil and thymol totally prevented proliferation. All growth was inhibited by bay oil, citronellal, and clove oil at 1000 ppm for 168 h. At 10,000 ppm, camphor prevented any growth for 168 h, whereas α-terpinene prevented any growth for 72 h but not for 168 h. In general, at concentrations below the threshold values, the degree of inhibition was dose-dependent [161]. Dellacasa et al. examined the fungicidal potential of eight oils in the *A. apis* vegetative cycle in 2003. They show that fungicidal action was present in varied degrees in the oils of *Tessaria absinthioides*, *Aloysia gratissima*, *Heterotheca latifolia, Lippia juneliana, L. integrifolia*, and *L. turbinata* but was absent in the oils of *Baccharis coridifolia* and *Eupatorium patens* [162]. In the same year, Davis and Ward, 2003, found *Leptospermum petersonii* F.M. Bailey (lemon-scented tea tree), *Eucalyptus citriodora* (Hook.) K.D. Hill and L.A.S. Johnson (lemon-scented Eucalyptus), *Leptospermum scoparium* J.R. Forst and G. Forst (manuka) to be effective in controlling *A. apis* [163]. Ruffinengo et al., 2006, discovered that *Heterothalamus alienus* oil-impregnated filter paper disks placed around colonies considerably inhibited the growth of *A. apis* by 51% compared to the control in a first trial and by 31% in a second [164]. In vitro growth of *A. apis* was also effectively inhibited by cinnamon, cloves, rose, thyme, and propolis, according to research by Abou El-Enain et al., 2009. Fennel, ginger, henna, onion, and wormwood oils, on the other hand, showed little inhibition in vitro. The pathogen’s development was not inhibited by amalaki, fenugreek, violet oils, or fennel honey plants in the same investigation [165]. Recently, the ability of EOs of orange, lemongrass, citronella, and cinnamon bark to control *A. apis* in both contact-dependent and contact-independent shared airspaces was investigated. All of the substances were discovered to considerably reduce mycelial growth at low concentrations, indicating the possibility of using these natural products to manage this and other specific fungal diseases [166]. Nardoni et al., 2018, verified that the dilutions tested of *C. zeylanicum* had no effect against *A. apis;* when assayed undiluted *C. zeylanicum* proved effective against *A. apis* [7,166,167]. Due to the high citral content of *Litsea cubeba* EO, monoterpene citral, which was also found in *C. flexuosus* EO, it looked to be quite effective in controlling *A. apis* [7]. According to reports, EOs with high geraniol concentration are active. *Pelargonium graveolens’* antifungal properties later supported this conclusion [7]. Kloucek et al., 2012, examined the effectiveness of a large number of EOs, specifically 70, in the vapour phase for the treatment of disease caused by *A. apis*. The researchers found that 39 of the 70 EOs had an antifungal impact on *A. apis*. The EO vapours of *Armoracia rusticana* exhibited the strongest antifungal activity, followed by *Thymus vulgaris*, *Cymbopogon flexosus*, *Origanum vulgare*, and *Allium sativum*. Allyl isothiocyanate, citral, carvacrol, and diallyl sulphides were the major components of the most active EOs [168]. Pusceddu et al., 2021 evaluated the efficacy of several EOs against *A. apis* as well. *Thymus herba-barona*, *Thymus capitatus*, and *Cinnamomum zeylanicum* were the most efficacious EOs with the lowest fungicidal and sporicidal concentrations within 200 and 400 ppm. *Thymus capitatus* and *Thymus herba-barona*’s oils were mostly composed of carvacrol, whereas *Cinnamomum zeylanicum*’s oil was predominately composed of cinnamic aldehyde [169]. There were discrepant findings with *S. aromaticum* [7,168]. Instead, when combined, certain potent EOs, such as *C. zeylanicum*, *C. flexuosus*, *P. graveolens*, and *L. cubeba* (0.02% each and 0.015% each), showed a synergistic impact that significantly increased their efficacy against *A. apis* [7].

### 3.4. Viruses

Although the direct harm that *V. destructor* inflicts on larvae and adult honeybees by sucking haemolymph is unquestionably significant, it pales in comparison to the transfer of viruses, which is the primary reason for colony breakdown and death. Over time, it has become clearer that viruses have a function in the presence of *V. destructor* on hive mortality. Over 20 viruses have been detected thus far. The deformed wing virus (DWV) is now known to be the most dangerous. On mature honeybees, this virus may be present naturally and seldom causes issues. Adult honeybees with malformed wings are created when the virus is directly injected into the haemolymph of the larvae by the *Varroa* parasite. The ability of DWV to reproduce in the mite may account for the tremendous effectiveness of *V. destructor* in spreading this virus; in fact, the virus spreads widely even in areas with low infestation levels [170]. It has also been proposed that the occurrence is linked to an immunodepression condition in growing host honeybees, which would also better explain the interaction between the virus and the mites. The lifespan of the honeybees will be shortened if acute paralysis virus (ABPV), which is often innocuous in the absence of *V. destructor*, is also present at the same time. This is the reason why colonies perish in the winter yet often even the most badly afflicted colonies survive in the summer because they can produce honeybees to replace those who die too soon. Additionally, *Varroa*-infested colonies are significantly impacted by the Israeli acute paralysis virus (IAPV). Recently, it has been shown that this virus has detrimental effects on foragers’ behaviour, specifically on their capacity to returning to the hive (homing ability) [171,172,173]. According to experts, the IAPV virus infects a bee’s neurological system and prevents it from returning to the hive [172]. Since it may multiply in *V. destructor* and is sent back to the honeybees from them, its spread is correlated with the severity of the infection. It appears more common for honeybees to become sick with numerous viruses than just one, yet seeing an afflicted bee is often uncommon [174]. In order to prevent or at least lessen the danger of viral transmission, it is necessary to maintain a reasonable level of the *V. destructor* populations. In general, the risk of virus multiplication rises as *Varroa* populations and its feeding activities rise [170]. There are not many studies concerning the control of bee viruses by administering natural products. In a 2016 study by Aurori et al., the antiviral effectiveness of ethanolic extracts of leaves from *Laurus nobilis* on foraging honeybees that were naturally infected with the BQCV (black queen cell virus) was examined. Even at the lowest dosage examined (1 mg/mL), overall virus loads decreased after therapy. Significantly less viral replication was seen at higher extract concentrations (5 mg/mL) [175]. In a more recent study by Boncristiani et al., 2021, several compounds were studied to demonstrate their beneficial activities on DWV and VDV levels in honeybees [176]. Although not all of the items examined had consistent outcomes, the researchers did discover several that had an impact on viral loads. Interestingly, when honeybees were supplemented with NPs, raw cacao had a substantial impact on the amount of DWV and VDV. Chocolate is made from fermented, dried, and roasted cocoa beans, which are produced by the *Theobroma cacao* tree in tropical rainforests. Polyphenols, flavonols, and procyanidins are only a few of the many bioactive chemical substances found in cocoa beans [177,178]. Prior to roasting, many of these chemical components are present in greater quantities [179], However, this causes losses of up to 71% for the total phenolic compounds and 53–77% for the antioxidants [180,181]. The cocoa bean shells utilised in Boncristiani’s research were both unroasted and roasted [176]. With comparable 30-fold increases in viral loads, but in the reverse direction, DWV levels rose while VDV levels fell [176]. Citrus fruits like lemons and oranges, as well as several other fruits and vegetables, contain a major flavonoid called hesperidin [182]. Hesperidin greatly lowered VDV levels [176]. Instead, the others therapies that exhibited effects (chrysin, curcumin, limonene, tocopherol, tyrosine) showed no statistically significant variations of viral loads [176]. Another recent study examined the effects of grape marc powder (GPP) as a dietary supplement to strengthen honeybees with DWV-affected immune systems. According to the findings, GPP treatments improved honeybees’ ability to fight against DWV. When honeybees were infected with DWV and given GPP (doses of 0.5, 1, 2.5, and 5%), the viral load was substantially different from that of the inoculated honeybees who had not received the GPP supplementation. Furthermore, the expression of the Relish gene was much greater in honeybees given GPP compared to the infected control.

### 3.5. Varroa destructor

*V. destructor* parasitises adult honeybees and their larval forms. Particularly in the brood, *V. destructor* produces enormous damage, causing malformations and stunting the development of individuals [183]. The damage is due to parasitic action. *V. destructor* is in fact an ectoparasite that feeds mainly on the fat body of immature and adult honeybees [184]. In adults, it also produces behavioural alterations and reduced biological functions. All of this can quickly lead to colony collapse if action is not taken in a short time [185]. Various synthetic acaricide products have been created over the years, but mites have often developed resistance to them [186,187]. Synthetic products also accumulate in the food matrices of the hive, resulting in toxicity to honeybees and possible risks to the end consumer [188]. For this reason, NPs may be a promising alternative. For example, EOs have often demonstrated efficacy of action [189,190,191,192], and it is known that they are easily degraded in the environment without giving rise to accumulation phenomena [22]. Moreover, given their complex chemical composition, mite populations are unlikely to develop resistance to them [193]. The plant families Apiaceae, Asteraceae, Lamiaceae, Myrtaceae, Poaceae, and Verbenaceae have received the greatest attention in studies of EO activity. Pure EOs and isolated monoterpenes have both been used in tests against *V. destrucor*. The relationship of EOs with entomopathogenic fungi has instead been anticipated by specific association studies [194,195]. For each family that was researched, we provide a few studies along with the findings on their effectiveness below. In several independent laboratory studies, the acaricidal abilities of *Syzygium aromaticum* were examined for the family Myrtaceae. When used as a fumigant, *S. aromaticum* demonstrated a high degree of heterogeneity in effectiveness. Similar but much better results were obtained by Sammataro et al., 1998 [196], and Vieira et al., 2012 [197], compared to Xiao-Ling et al., 2012 [198]. While the second study group’s average mortality was 54%, the first two research groups’ average death was over 87%. *Citrus* spp. and *Mentha* spp. EOs both displayed comparable non-constant acaricidal efficacy. Acaricidal effectiveness from isolated EOs was shown to be excellent in some trials but not in others [8,197,199,200,201]. In laboratory investigations, plants from the species *Pimpinella* spp. and *Foeniculum* spp. of the Apiaceae family were assessed. *Foeniculum* spp. had a lower acaricidal capability, typically about 60% [37,189], compared to *Pimpinella* spp., which was reported to have an effectiveness of 92.5% in studies by Vieira et al., 2012 [197], and Xiao-Ling et al., 2012 [198]. Plants from the groups Verbenaceae, Lauraceae, and Poaceae have received little attention in research. The species *Acantholippia seriphioides* (aerial parts) of the Verbenaceae family was tested by Ruffinengo et al. in 2014 [202], and it showed a high acaricide efficiency of 99% via complete exposure and 87% via fumigation. The species *Laurus nobilis* [196] and *Cinnamomum verum* [197] were primarily investigated for the Lauraceae family. Particularly, Vieira et al., 2012, [197] discovered that *Cinnamomum verum* fumigation had only 52.50% acaricidal activity after 6 h of fume exposure. The same botanical family member, *Laurus nobilis*, produced superior results, achieving an acaricidal efficacy close to 75%. For the Lamiaceae family, in addition to thyme, which demonstrated consistently high acaricidal efficacy results in independent studies, oregano essential oil has also shown great promise [12,39,203,204].

Table 1 below summarises the botanical species object of the studies mentioned in the manuscript.

## 4. Discussion and Conclusions

In addition to summarising the NPs that have been reported in the literature for their positive effects for the control of hive pathogens, this scientific paper reviews the current state of research on the mechanism of action of EOs. They are already available in the beekeeper’s kit for controlling the *V. destructor* parasite. In particular, the pharmacological preparations on the market are based on thymol. In this literature review, it was observed that thymol is also effective in the control of bacterial, fungal, and viral infections. In addition to thymol, several other compounds have shown promise for pathogen control. These NPs could be used more systematically in the future. However, some obstacles must be overcome for their use. Natural compounds have different chemical compositions depending on the period and place of the harvesting of the plants from which they are extracted. Furthermore, NPs have chemical structures and numerous chiral centres that make their synthesis difficult. When isolated directly from their main biological resource, the extraction method can influence the presence of certain molecules. The lack of standardised criteria and standards in the present research on NPs might result in underdeveloped methods for assessing their efficacy and possibly confusing procedures. Another issue is that only small quantities of products are often isolated. The evaluation of compounds that can only be obtained in small quantities often takes place with difficulty and over a long time. Miniaturised bioassays have been created to overcome this problem. Large-scale research should be conducted to examine the interactions between dietary supplements and their antagonists. In reality, we have found that there are not many association studies. Additionally, when exposed to oxidizing environmental factors like sun and air, natural chemicals are very labile. Therefore, it is vital to use technologies that safeguard preparations. We may state as a conclusion that further study is required to address the challenges associated with the usage of NPs.

## Figures and Tables

**Figure 1 pathogens-12-01260-f001:**
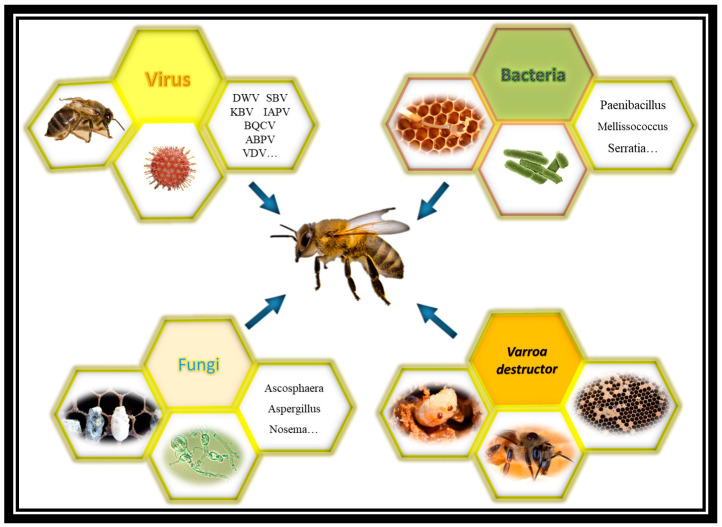
Major pathogens that affect honeybees. The primary pathogens that infect *Apis mellifera* are shown schematically as members of the taxonomic grouping of fungi, bacteria, and viruses. An area of the image has been entirely dedicated to *V. destructor* for its important role as a vector of pathogens.

**Table 1 pathogens-12-01260-t001:** Botanical species used for pathogens control in honeybee and their targets.

Botanical Name	Common Name	Activity	References
*Acantholippia seriphioides*	Acantholippia	*V. destructor* *Paenibacillus larvae*	[36,126,202]
*Achyrocline satureioides*	Macela	*Paenibacillus larvae*	[130,131,132]
*Allium sativum*	Garlic	*Ascosphaera apis*	[168]
*Aloysia gratissima*	Beebrush, Whitebrush	*Paenibacillus larvae*	[162]
*Aloysia polystachya*	Beebrush	*Paenibacillus larvae*	[126]
*Aristotelia chilensis*	Chilean wineberry	*Nosema* spp.	[153]
*Armoracia rusticana*	Horseradish	*Paenibacillus larvae* *Ascosphaera apis*	[122,168]
*Artemisia dubia*	Mugwort	*Nosema* spp.	[146]
*Aster scaber*	*Chwinamul*	*Nosema* spp.	[146]
*Azadirachta indica*	Neem	*Paenibacillus larvae*	[129]
*Baccharis coridifolia*	Mio-mio, Vassourinha	*Ascosphaera apis*	[162]
*Baccharis latifolia*	Chilca	*Paenibacillus larvae*	[126]
*Brassica nigra*	Black mustard	*Nosema* spp.	[152]
*Buddleja globosa*	Orange-ball-tree, orange ball	*Paenibacillus larvae*	[126]
*Calendula officinalis*	Pot marigold	*Paenibacillus larvae*	[135]
*Camellia sinensis*	Tea plant, tea bush	*Nosema* spp.	[148]
*Chenopodium ambrosioides*	Mexican tea	*Paenibacillus larvae*	[130]
*Cinnamomum* sp.	Cinnamon	*Ascosphaera apis*	[166]
*Cinnamomum zeylanicum*	Dalchini	*Melissococcus plutonius* *Paenibacillus larvae Ascosphaera apis*	[7,137,169]
*Citrus reticulata*	Mandarin orange	*Ascosphaera apis*	[165]
*Citrus* spp. *(hesperidin)*	Citrus	*V. destructor* Deformed wing virus (DWV)	[8,176]
*Coriander sativum*	Cilantro, Chinese parsley	*Nosema* spp.	[145]
*Cryptocaria alba*	Peumo	*Nosema* spp.	[144]
*Cymbopogon citratus*	Lemongrass	*Paenibacillus larva; Ascosphaera apis*	[35]
*Cymbopogon flexosus*	Lemongrass	*Ascosphaera apis*	[168]
*Eleutherococcus senticosus*	*Siberian ginseng,* eleuthero	*Nosema* spp.	[148]
*Eruca sativa*	Rocket leaves	*Nosema* spp.	[152]
*Eucalyptus cinerea*	Argyle apple	*Paenibacillus larvae*	[130]
*Eucalyptus citriodora*	Lemon-scented gum	*Ascosphaera apis*	[163]
*Eucalyptus globulus*	Tasmanian bluegum	*Paenibacillus larvae*	[35]
*Eupatorium patens*	Boneset	*Ascosphaera apis*	[162]
*Flourensia fiebrigii*	Chilca, maravilla	*Paenibacillus larvae*	[127]
*Flourensia riparia*	Riparian Flourensia	*Paenibacillus larvae*	[127]
*Flourensia tortuosa*	Flourensia	*Paenibacillus larvae*	[127]
*Foeniculum* spp.	Fennel	*V. destructor*	[37]
*Foeniculum vulgare*	Fennel	*Ascosphaera apis*	[165]
*Garcinia cambogia*	*Gambooge, Malabar Tamarind*	*Nosema* spp.	[148]
*Gevuina avellana*	Chilean wildnut	*Nosema* spp.	[153]
*Gnaphalium gaudichaudianum*	Pseudognaphalium	*Paenibacillus larvae*	[130]
*Heterothalamus alienus*	Romerillo	*Ascosphaera apis*	[164]
*Heterotheca latifolia*	Camphorweed	*Ascosphaera apis*	[162]
*Hypericum* spp.	Common St. John’s wort	*Paenibacillus larvae*	[128]
*Laurus nobilis*	Bay laurel	*V. destructor* *Paenibacillus larvae* Black queen cell virus (BQCV)	[117,175,196]
*Lavandula hybrida*	Lavandin	*Paenibacillus larvae*	[35]
*Leptospermum petersonii*	Lemon-scented tea-tree	*Ascosphaera apis*	[163]
*Leptospermum scoparium*	Tea-tree	*Ascosphaera apis*	[163]
*Lippia integrifolia*	Hieron	*Ascosphaera apis*	[162]
*Lippia juneliana*	Lippia junelia	*Ascosphaera apis*	[162]
*Lippia turbinata*	Lippia turbinata	*Paenibacillus larvae; Ascosphaera apis*	[126,130,162]
*Litsea cubeba*	Mountain pepper	*Ascosphaera apis*	[7]
*Marrubium vulgare*	White horehound	*Paenibacillus larvae*	[130]
*Melia azedarach*	Chinaberry tree	*Paenibacillus larvae*	[133]
*Melissa officinalis*	Lemon balm	*Nosema* spp.	[145]
*Mentha piperita*	Peppermint	*Paenibacillus larvae* *Nosema* spp.	[35,145]
*Mentha spicata* var. *crispa*	Peppermint	*Paenibacillus larvae*	[122]
*Minthostachys mollis*	Muña, Peperina	*Paenibacillus larvae*	[126]
*Minthostachys verticillata*	Peperina	*Paenibacillus larvae*	[130]
*Nasturtium officinale*	Watercress	*Paenibacillus larvae*	[135]
*Origanum* spp.	Oregano	*V. destructor*	[12,204]
*Origanum vulgare*	Oregano	*Paenibacillus larvae* *Ascosphaera apis*	[35,125,130,168]
*Pelargonium graveolens*	Rose geranium	*Ascosphaera apis*	[7]
*Pimenta racemosa*	Bay rum tree	*Ascosphaera apis*	[161]
*Polygonum bistorta*	Meadow bistort	*Melissococcus plutonius* *Paenibacillus larvae*	[124]
Salvia Rosmarinus	Rosemary	*Paenibacillus larvae*	[35]
*Satureja hortensis*	Cibru, savory	*Paenibacillus larvae* *Nosema* spp.	[35,122,145]
*Schinus molle*	Peruvian peppertree	*Paenibacillus larvae* *Nosema* spp.	[126]
*Schisandra chinensis*	Chinese magnolia vine	*Nosema* spp.	[148]
*Scutia buxifolia*	Boxleaf *scutia*	*Paenibacillus larvae*	[134]
*Solidago chilensis*	Goldenrod	*Paenibacillus larvae*	[126]
*Syzygium aromaticum*	Clove	*V. destructor* *Paenibacillus larvae* *Ascosphaera apis* *Melissococcus plutonius*	[125,137,161,196,197]
*Tagetes minuta*	Stinking Roger	*Paenibacillus larvae*	[130]
*Tessaria absinthioides*	Tessaria	*Ascosphaera apis*	[162]
*Theobroma cacao*	Cocoa, cacao	Deformed wing virus (DWV)	[176]
*Thymus capitatus*	Conehead thyme	*Ascosphaera apis*	[161,169]
*Thymus herba-barona*	Caraway thyme	*Ascosphaera apis*	[169]
*Thymus vulgaris*	German thyme	*Melissococcus plutonius* *Paenibacillus larvae* *Ascosphaera apis*	[35,122,125,130,137,168]
*Ugni molinae*	Chilean guava, strawberry myrtle	*Nosema* spp.	[153]
*Chrysopogon zizanioides*	Vetiver	*Nosema* spp.	[142]
*Vitex trifolia*	Simpleleaf chastetree	*Paenibacillus larvae*	[129]

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
