# Peer review of "Plants and Their Derivatives as Promising Therapeutics for Sustainable Control of Honeybee (Apis mellifera) Pathogens"

_pathogens, 2023, doi:10.3390/pathogens12101260_

Round 1

Reviewer 1 Report

The authors highlight the scientific community's progress over the years in employing natural substances to treat honey bee diseases. The review is highly informative and well written. I had only few comments and suggestions:

1- Given the severe consequences of varroa mite infestation, I would suggest that the authors include a section on varroa mites and the role of EO in their control.

2-After section 2.3, I missed a transition paragraph introducing the most important and researched honey bee pathogens and pests.

Minor comments:

1- line 219 and throughout the review: illness should be replaced by disease as illness most refers to patients.

2- line 366: eucariotic should be corrected to eukaryotic

3- line 474: Due to not Due of.

4- lines 532 and 545: family should be replaced by colony.

Author Response

The authors highlight the scientific community's progress over the years in employing natural substances to treat honey bee diseases. The review is highly informative and well written. I had only few comments and suggestions:

R.: Many thanks to the reviewer for his advice and critical reading, which improved the overall quality of the manuscript. We have made changes by highlighting them in the text.

1- Given the severe consequences of varroa mite infestation, I would suggest that the authors include a section on varroa mites and the role of EO in their control.

R.: Two new paragraph on Varroa destructor have been added as recommended (the first in the section “mechanism of action”, the second one is a dedicated part to the parasite and its control by natural products).

2-After section 2.3, I missed a transition paragraph introducing the most important and researched honey bee pathogens and pests.

R.: Thanks for the suggestion. A short paragraph introducing and transitioning to the most important pathogens has been added.

Minor comments:

1- line 219 and throughout the review: illness should be replaced by disease as illness most refers to patients.

R.: many thanks for suggestion. The word has been changed throughout the text as suggested

2- line 366: eucariotic should be corrected to eukaryotic

R.: Now amended

3- line 474: Due to not Due of.

R.: Now amended

4- lines 532 and 545: family should be replaced by colony.

R.: Now amended

Reviewer 2 Report

General comments

- The manuscript is an interesting Review of plants and their derivatives as promising therapeutics for controlling honey bee diseases.

- Line 163, 208 and 255: Rosmarinus officinalis as far as I know, is now named Salvia Rosmarinus

- Line 243: In the paragraph referring to AFB control techniques, I would suggest adding the use of the shaking method where the combs are removed and burned, but the colony can survive and heal via the construction of new combs

- Line 370-372: the authors refer to Nosema species, but I believe they have been confused regarding the symptoms. Nosema apis causes typical digestive symptoms, while Nosema ceranae has no particular signs.

- Line 523-525: The statement that thymol affects insect behavior, thus is not recommended for use in beekeeping is contrary to the general approach of the manuscript where there are many references to the use of thymol. Even in the conclusions (Line 589-590) the authors declare, “preparations on the market are based on thymol, which is effective in the control of bacteria, fungal and viral infections”.

- Line 583: Table 1 has no reference in the manuscript body.

Some necessary corrections to my opinion:

Line 49: “honeybees colonies” change to “honeybee colonies”

Line 52: “illnesses” change to “diseases”

Line 71: “so far mostly been” change to “so far been mostly”

Line 79: “special” change to “particular”

Line 206: correct “of the viral life”

Line 219: correct “AFB is the most”

Line 224: correct “bacteria originated”

Line 252: “are below presented” change to “are presented below”

Line 256: correct “the strongest action was found”

Line 326: “But” change to “However”

Line 356: “effectiveness” change to “efficacy”

Line 371: “diarrhea” change to “diarrhoea”

Line 474: “Due of” change to “Due to”

Line 476: “at danger” change to “in danger”

Line 575: correct “showed no statistically”

Line 602: “in a long time” change to “over a long time”

Line 605: Remove “out there”

Minor editing of English language required

Author Response

General comments

- The manuscript is an interesting Review of plants and their derivatives as promising therapeutics for controlling honey bee diseases.

R.: We sincerely thank the reviewer for his suggestions, which enabled us to improve the overall quality of the paper; the changes indicated have been made and highlighted in the text.

- Line 163, 208 and 255: Rosmarinus officinalis as far as I know, is now named Salvia Rosmarinus

R.: Now amended

- Line 243: In the paragraph referring to AFB control techniques, I would suggest adding the use of the shaking method where the combs are removed and burned, but the colony can survive and heal via the construction of new combs

R.: Thank you very much for this suggestion. A sentence regarding this pathology control technique has been added.

- Line 370-372: the authors refer to Nosema species, but I believe they have been confused regarding the symptoms. Nosema apis causes typical digestive symptoms, while Nosema ceranae has no particular signs.

R.: Many thanks to the reviewer for pointing out this typo. We have made the changes by correctly relating the disease symptoms to each species.

- Line 523-525: The statement that thymol affects insect behavior, thus is not recommended for use in beekeeping is contrary to the general approach of the manuscript where there are many references to the use of thymol. Even in the conclusions (Line 589-590) the authors declare, “preparations on the market are based on thymol, which is effective in the control of bacteria, fungal and viral infections”.

R.: We thank the reviewer for his comment. The information we reported on thymol seemed important to us, although contradictory to the general spirit of the manuscript. In light of the suggestion made, we agree that it should be removed from the final version.

- Line 583: Table 1 has no reference in the manuscript body.

R.: Many thanks for this comment. A reference in the text has been added as consequence

Some necessary corrections to my opinion:

Line 49: “honeybees colonies” change to “honeybee colonies”

R.: Now amended

Line 52: “illnesses” change to “diseases”

R.: Now amended

Line 71: “so far mostly been” change to “so far been mostly”

R.: Now amended

Line 79: “special” change to “particular”

R.: Now amended

Line 206: correct “of the viral life”

R.: Now amended

Line 219: correct “AFB is the most”

R.: Now amended

Line 224: correct “bacteria originated”

R.: Now amended

Line 252: “are below presented” change to “are presented below”

R.: Now amended

Line 256: correct “the strongest action was found”

R.: Now amended

Line 326: “But” change to “However”

R.: Now amended

Line 356: “effectiveness” change to “efficacy”

R.: Now amended

Line 371: “diarrhea” change to “diarrhoea”

R.: Now amended

Line 474: “Due of” change to “Due to”

R.: Now amended

Line 476: “at danger” change to “in danger”

R.: Now amended

Line 575: correct “showed no statistically”

R.: Now amended

Line 602: “in a long time” change to “over a long time”

R.: Now amended

Line 605: Remove “out there”

R.: Now amended

Reviewer 3 Report

From the complexity point of view of the review, it would be appropriate to include also varroa, although I agree with autors that recent reviews (but also new findings) exists on this topic (f.e. doi: 10.3390/vetsci10050308).

The work provides a summary view of this issue and in general, I have no fundamental comments on its concept, although certainly there would still be studies that were not included to this review.

It is necessary to check the formal uniformity of the text, e.g. writing Latin names in italics (f.e. in lines 162, 165, 167), cited authors (f.e. 109 – 110), full names of plants (f.e. in lines 319 – 334, 430, 445, 504, 518 - 520, 525 – 527).

Chapter 2: text in lines 169 – 177, as well as 211 – 217 doesn't quite fit with the title of the chapter (Mechanisms of action) or could be moved to subsequent chapters.

Chapter 3: several genotypes of Paenibacillus larvae with variable infection importance exists. Maybye will be better to include in this Chapter also other bacterial agents (EFB), as data on EOs impact also exists (f.e. doi: 10.3390/antibiotics10080960). The description of the experiment on lines 264 – 277 could be condensed.

Chapter 4: rephrase text on lines 387 -389. There are much more feeding supplements with claims to help fight against Nosema.

Chapter 5: also data on use of Andrographis paniculata EO exists (Davis, C., & Ward, W., 2003).

Chapter 6: also work on BQCV, doi: 10.1016/j.virusres.2016.05.024 should be mentioned.

Author Response

From the complexity point of view of the review, it would be appropriate to include also varroa, although I agree with authors that recent reviews (but also new findings) exists on this topic (f.e. doi: 10.3390/vetsci10050308).

R.: We thank the reviewer for his suggestion which helps to make the article more complete. We have added a paragraph on Varroa destructor and the treatments with natural products that have been tested against it over time. In addition, a paragraph on the mechanisms of acaricidal activity of natural products has been added.

The work provides a summary view of this issue and in general, I have no fundamental comments on its concept, although certainly there would still be studies that were not included to this review.

R.: Thank you very much for your comment. More studies have been added for each disease as suggested

It is necessary to check the formal uniformity of the text, e.g. writing Latin names in italics (f.e. in lines 162, 165, 167), cited authors (f.e. 109 – 110), full names of plants (f.e. in lines 319 – 334, 430, 445, 504, 518 - 520, 525 – 527).

R.: The entire text was analysed and corrected for italic where necessary.

Chapter 2: text in lines 169 – 177, as well as 211 – 217 doesn't quite fit with the title of the chapter (Mechanisms of action) or could be moved to subsequent chapters.

R.: Thanks to the reviewer for his comment. The text in lines 169 – 177 has been deleted. In the case of viruses and their control with natural products, there are few studies concerning honeybees. Although viruses belonging to other species are mentioned in lines 211 to 217, with the reviewer's approval we would like to keep these few lines to emphasise the possibility of controlling viruses using essential oils.

Chapter 3: several genotypes of Paenibacillus larvae with variable infection importance exists. Maybye will be better to include in this Chapter also other bacterial agents (EFB), as data on EOs impact also exists (f.e. doi: 10.3390/antibiotics10080960). The description of the experiment on lines 264 – 277 could be condensed.

R.: Thank you very much to the reviewer, whose comments enable us to provide more completeness to the paper. Information on studies concerning EFB has been added, and, as suggested, the experiment included between lines 264 and 277 has been described more concisely.

Chapter 4: rephrase text on lines 387 -389. There are much more feeding supplements with claims to help fight against Nosema.

R.: Thank you for this comment. The text has been reworded.

Chapter 5: also data on use of Andrographis paniculata EO exists (Davis, C., & Ward, W., 2003).

R.: Many thanks for this comment. The suggested study was added

Chapter 6: also work on BQCV, doi: 10.1016/j.virusres.2016.05.024 should be mentioned.

R.: Many thanks for this advice. The suggested study was added

Round 2

Reviewer 1 Report

The authors did a good job in revision and addressed my comments.